# Uncertainty in the Number of Calibration Repetitions of a Hydrologic Model in Varying Climatic Conditions

**Patrik Sleziak [1],\* , Ladislav Holko [1], Michal Danko [1] and Juraj Parajka [2]**

[1]  Institute of Hydrology, Slovak Academy of Sciences, Dúbravská cesta 9, 84104 Bratislava, Slovakia; holko@uh.savba.sk (L.H.); danko@uh.savba.sk (M.D.)

[2]  Institute of Hydraulic Engineering and Water Resources Management, Technische Universität Wien, Karlsplatz 13/223, 1040 Vienna, Austria; parajka@hydro.tuwien.ac.at

\*  Correspondence: sleziak@uh.savba.sk; Tel.: +421-908-965-784

**Abstract:** The objective of this study is to examine the impact of the number of calibration repetitions on hydrologic model performance and parameter uncertainty in varying climatic conditions. The study is performed in a pristine alpine catchment in the Western Tatra Mountains (the Jalovecký Creek catchment, Slovakia) using daily data from the period 1989–2018. The entire data set has been divided into five 6-years long periods; the division was based on the wavelet analysis of precipitation, air temperature and runoff data. A lumped conceptual hydrologic model TUW ("Technische Universität Wien") was calibrated by an automatic optimisation using the differential evolution algorithm approach. To test the effect of the number of calibrations in the optimisation procedure, we have conducted 10, 50, 100, 300, 500 repetitions of calibrations in each period and validated them against selected runoff and snow-related model efficiency criteria. The results showed that while the medians of different groups of calibration repetitions were similar, the ranges (max–min) of model efficiency criteria and parameter values differed. An increasing number of calibration repetitions tend to increase the ranges of model efficiency criteria during model validation, particularly for the runoff volume error and snow error, which were not directly used in model calibration. Comparison of model efficiencies in climate conditions that varied among the five periods documented changes in model performance in different periods but the difference between 10 and 500 calibration repetitions did not change much between the selected time periods. The results suggest that ten repetitions of model calibrations provided the same median of model efficiency criteria as a greater number of calibration repetitions and model parameter variability and uncertainty were smaller.

**Keywords:** hydrological model uncertainties; optimisation of model parameters; climate change

---

## 1. Introduction

Conceptual rainfall-runoff models are used for a wide range of purposes including reservoir operations, flood and drought prediction, risk analysis, climate change impact studies, etc. [1,2]. These models usually contain parameters that need to be estimated through calibration. However, many parameters sets with different values of the same parameters can provide similar results in terms of model efficiency, which is termed as "the equifinality principle" [3,4]. The problem of equifinality and its reduction in hydrological modelling has been discussed in many studies (e.g., [3–7]). Previous investigations examined the potential of the Monte Carlo approach, multiple-objective calibration or the influence of the length of data series or climatic variability during calibration/validation periods on the improvement of parameter representativeness and hence reduction of uncertainty in parameters of the models. Several authors (e.g., [8–11]) used the Monte Carlo (MC) calibration approach to examine

the links between model consistency and parameter uncertainty. For example, Finger et al. [9] and [10] used the MC approach (10,000 runs) to test the performance of a conceptual hydrologic model in terms of three observational data sets (discharge, snow cover, glacier mass balances) and found that an ensemble of 100 parameters sets adequately represented parameter variability, hydrological regime and seasonal dynamics of discharge. Another approach for selection of representative model parameters was recently evaluated by Sikorska-Senoner et al. [12] who tested three methods (ranking, quantiling, clustering) and found that the reduced parameter ensembles (i.e., three representative parameter sets) obtained by these methods were reliable for the simulation of extreme floods.

The multi-criteria approach, i.e., consideration of additional hydrological processes in model performance evaluation, has been successful in the reduction of parameter uncertainty, particularly for parameters connected to the additional process. For example, Finger et al. [9] and Parajka and Blöschl [13] showed that the combination of discharge data with snow cover related observations was useful in constraining snow model parameters. Similar results were obtained with satellite data of snow cover [11,14,15] or soil moisture [16].

Evaluations of the effect of length of calibration period on model parameter uncertainty indicated that longer calibration periods reduced parameter uncertainty [17]. Generally, longer calibration periods are advised to capture the variability of climatic and flow conditions [18]. Anctil et al. [19] and Brath et al. [20] recommended using two to ten years as optimum for model calibration. Merz et al. [21] stated that since the calibration period of five years captured most of the temporal hydrological variability, it should be the minimum for achieving reasonable predictive model performance.

A number of studies analysed the effects of climatic variability on the variability of model parameters (e.g., [22–27]). Several authors observed a decreasing trend in model performance when model parameters were transferred to periods with a different climate, e.g., from wetter to drier periods and vice versa [22,23,28–35]. For example, in Australia, Vaze et al. [28] and Coron et al. [23] found that the transfer of model parameters to a drier climate was problematic. They concluded that the model parameters transferability was more influenced by a change in precipitation than by changes in evaporation or air temperature. In Austria, Merz et al. [22] found that parameters representing snow cover evolution and soil moisture variability showed significant correlations with air temperature. They documented that a model calibrated in a colder/drier decade had a tendency to overestimate runoff in a warmer/wetter decade. Similar results were obtained by [34] who showed that the use of parameters calibrated in a colder decade for a warmer/wetter decade tends to overestimate catchment runoff, particularly in flatland and hilly catchments.

The uncertainties associated with hydrological model calibration (i.e., parameter estimation, choice of the length of calibration and validation periods) remain a challenge for the modellers. Representativeness of model parameters and reproducibility and repeatability of results are fundamental assumptions in any calibration experiment and uncertainty assessment. The study of Ceola et al. [36] showed that even if the same experimental protocol is used for calibration of hydrological models, there can be some variability in model performance and parameter uncertainty. The results of Ceola et al. [36] indicate that repetition of the calibration procedure can help to detect insensitive model parameters and reduce equifinality, but it is still not clear how to estimate the optimal number of repetitions. This is particularly interesting at present when the variability of climate can have a significant impact on the temporal stability of model parameters [22].

The main aim of this study is hence to examine how does a different number of calibration repetitions impact hydrologic model uncertainty in varying climatic conditions. We investigate the following research questions: (a) Does the number of calibrations influence model performance in varying climatic conditions? (b) Is the optimal number of calibrations related to the varying climate conditions? (c) Does the increasing number of calibrations decrease parameter uncertainty? These questions are investigated in a small pristine alpine catchment in the Western Tatra Mountains where runoff generation is affected only by natural processes. Model performance is evaluated by criteria related to catchment runoff snow cover.

## 2. Study Catchment and Data

### 2.1. Study Catchment

The Jalovecký Creek catchment in the Western Tatra Mountains (northern Slovakia) is selected as a pilot catchment for this research (Figure 1). It is a small experimental catchment where the Institute of Hydrology SAS has carried out hydrological research since 1986. It is located at altitudes between 800 and 2178 m a.s.l. (meters above sea level) (mean 1500 m a.s.l.) and has a total area of 22.2 km$^2$. The average catchment slope is 30°, and most slopes have a south-eastern orientation. Soils are represented by Cambisol, Podsol, Ranker and Lithosol. All soils have high stoniness (typically 40–50% and more, [37,38]). Forests (mainly spruce) cover 44% of the catchment area. Dwarf pine covers 31% and Alpine meadows and bare rocks cover the rest 25% of the catchment area.

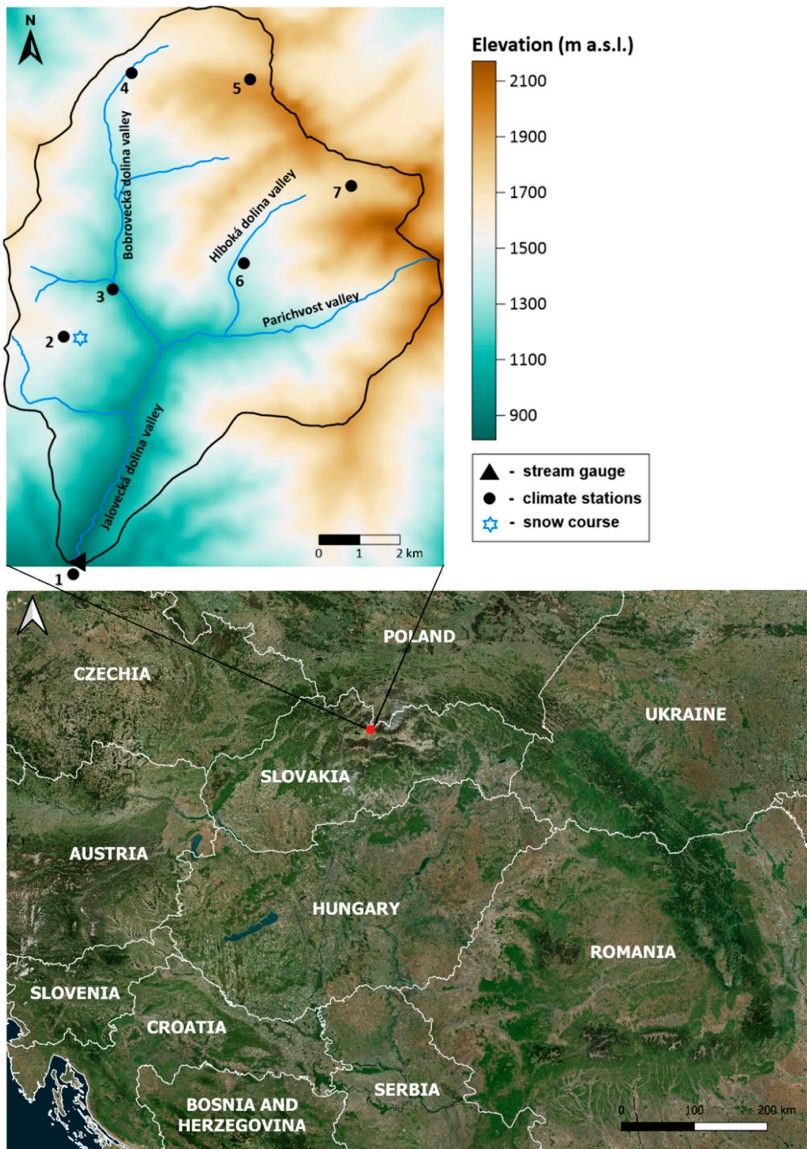

**Figure 1.** Map of Europe (**bottom**)—the red point indicates the location of the study catchment within Slovakia. The topography of the Jalovecký Creek catchment (**top**)—the black triangle indicates stream gauge; the black circles are climate stations providing the air temperature (numbers 1, 2 and 5) and precipitation data (numbers 1–7); the blue star represents site with snow water equivalent measurement.

## 2.2. Data

The study region is in a transitional zone between oceanic western and continental climate and belongs to the tundra (ET) climate class of Köppen classification. Climate observations, i.e., daily precipitation, daily air temperature, daily discharge and snow course measurements are available from the period between 1 November 1988 and 31 October 2018. Point observations of precipitation and air temperature are measured at 7 and 3 sites, respectively. Snow water equivalent is measured at site Červenec, which is located at catchment mean elevation (Figure 1).

## 2.3. Selection of Periods With Varying Climate

In this study, the entire dataset is divided into calibration/validation periods identified by the wavelet transform method. Based on the analysis of Sleziak et al. [39], we selected five 6-years long periods (i.e., P1 = 1989–1994, P2 = 1995–2000, P3 = 2001–2006, P4 = 2007–2012 and P5 = 2013–2018). The climatic and hydrologic characteristics of the selected periods are presented in Figure 2 and Table 1. The characteristics show that warmer periods (1989–1994 and 2013–2018) are generally drier. The largest difference in air temperature is observed between January and March. In the wettest period P2 (1995–2000), winters are characterized by larger snowpacks and mean an annual maximum of snow water equivalent is about 136 mm larger than in snow poor warmer periods. The largest differences in the seasonal distribution of precipitation and runoff are observed between April and September where the warmer periods tend to have less precipitation and runoff.

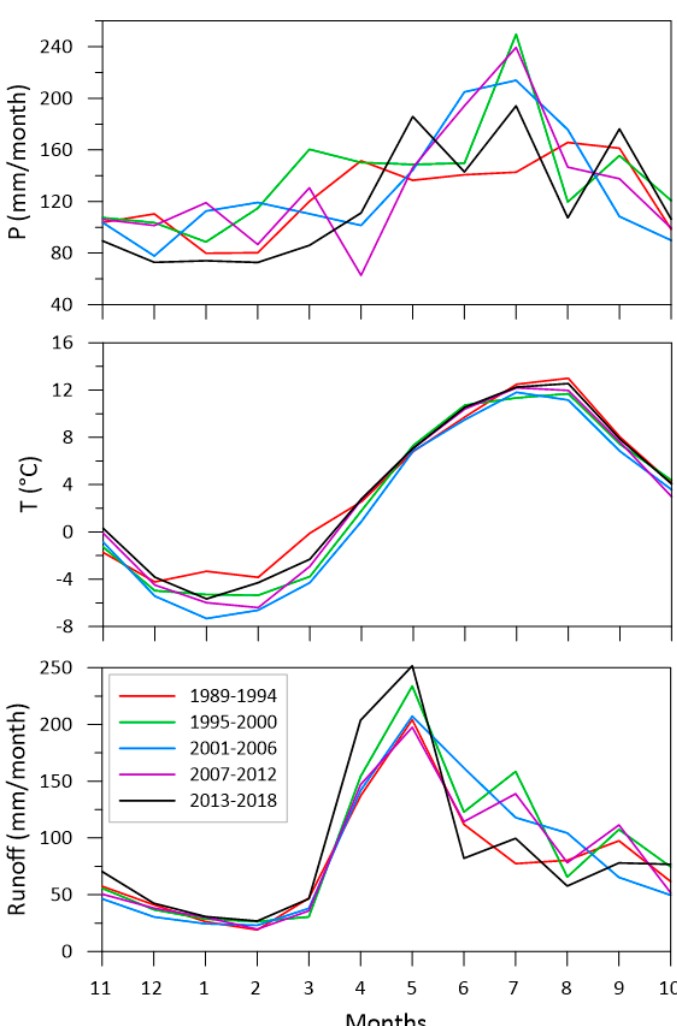

**Figure 2.** Variability of mean monthly precipitation, air temperature and runoff in five specific periods.

**Table 1.** Mean annual and seasonal characteristics of precipitation (P), air temperature (T), runoff (Q) and snow water equivalent (SWE) in the selected periods. Seasonal characteristics of P and T represent the period from April to October, SWEmax is the mean of SWE maximum in each year.

| | Annual Characteristics | | | Seasonal Characteristics | | |
|---|---|---|---|---|---|---|
| Period | P (mm/year) | T (°C) | Q (mm/year) | P (mm/) | T (°C) | SWEmax (mm) |
| 1989–1994 | 1491 | 3.6 | 973 | 997 | 8.1 | 284 |
| 1995–2000 | 1669 | 2.8 | 1109 | 1095 | 7.8 | 420 |
| 2001–2006 | 1563 | 2.1 | 1023 | 1039 | 7.3 | 408 |
| 2007–2012 | 1570 | 2.9 | 1027 | 1026 | 7.9 | 416 |
| 2013–2018 | 1419 | 3.4 | 1093 | 1024 | 8.2 | 344 |

## 3. Methods

### 3.1. Hydrologic Model

The hydrological model used in this study is the TUW model [40]. It is a conceptual Hydrologiska Byråns Vattenbalansavdelning (HBV) type model that uses daily precipitation totals, mean daily air temperatures, and potential evapotranspiration as the inputs. Mean daily flows observed at the outlet of the catchment and point measurements of snow water equivalent (Figure 1) are used for the evaluation of model simulations.

The structure of the model involves three modules that simulate changes in snow, soil water and groundwater storage. In total, the model has 15 parameters (Table 2). The snow module represents snow accumulation and melting in a catchment. Snowmelt is simulated by the degree-day approach, using a degree-day factor DDF (mm/°C/d) and a melt air temperature parameter Tm (°C). The catch deficit of precipitation gauges during snowfall is corrected by a snow correction factor SCF (–). A threshold temperature interval Tr − Ts (°C) is used to discriminate between rainfall, snowfall and mix of rain and snow [41]. The soil module simulates the processes taking place in the soil profile. It contains three parameters: the field capacity FC (mm), the limit for potential evapotranspiration Lprat (-), and the parameter relating runoff generation to the soil moisture state, termed the nonlinearity parameter BETA (–). If BETA is large, direct runoff is small, and vice versa. Finally, the runoff module consists of two reservoirs that represent hillslope routing. Rainfall enters the upper reservoir and leaves it through three paths: (1) outflow from the reservoir based on a very fast storage coefficient k0 (days) if a threshold of the storage state LSuz (mm) is exceeded in the upper reservoir; (2) outflow from the upper reservoir with a fast storage coefficient k1 (days), and; (3) percolation to the lower reservoir with a constant percolation rate Cperc (mm/day). Water leaves the lower zone based on a slow storage coefficient k2 (days). Channel routing is simulated by a triangular weighting function, where Bmax (days) is the maximum base at low flows, and Croute (day$^2$/mm) is a free scaling parameter. A detailed description of the model structure with particular model equations is given in Parajka et al. [41].

The model is automatically calibrated using a differential evolution algorithm Deoptim [42]. The objective function (OF) used in calibration is selected on the basis of prior analyses performed in different calibration studies [34,39]. It consists of a combination of the Nash-Sutcliffe coefficient (NSE, [43]) and the logarithmic Nash-Sutcliffe coefficient (logNSE, [22]):

$$OF = \frac{1 - NSE}{2} + \frac{1 - \log NSE}{2} \quad (1)$$

where NSE and logNSE criteria are mathematically expressed as follows:

$$NSE = 1 - \frac{\sum_{i=1}^{n} \left( Q_{sim,i} - Q_{obs,i} \right)^2}{\sum_{i=1}^{n} \left( Q_{obs,i} - \overline{Q}_{obs} \right)^2} \quad (2)$$

$$\log NSE = 1 - \frac{\sum\limits_{i=1}^{n}\left(\log(Q_{sim,i}) - \log(Q_{obs,i})\right)^2}{\sum\limits_{i=1}^{n}\left(\log(Q_{obs,i}) - \log(\overline{Q}_{obs})\right)^2} \tag{3}$$

$Q_{sim,i}$ and $Q_{obs,i}$ indicate the simulated and observed mean daily flows on day $i$, and $\overline{Q}_{obs}$ is the average of the flows observed. The NSE and logNSE coefficients range between $-\infty$ (poor fit) and 1 (perfect fit of the observed and simulated values).

**Table 2.** 15 TUW model parameters according to the routines and their calibration ranges. The calibration ranges were taken from the literature [22].

| Parameter | Routine | Unit | Range |
|---|---|---|---|
| Snow correction factor (SCF) | Snow | | 0.9–1.5 |
| Degree-day factor (DDF) | Snow | mm/°C day | 0–5 |
| Rain threshold temperature (Tr) | Snow | °C | 1–3 |
| Snow threshold temperature (Ts) | Snow | °C | −3–1 |
| Melt temperature (Tm) | Snow | °C | −2–2 |
| Limit for potential evapotranspiration (Lprat) | Soil | day | 0–l |
| Maximum soil moisture storage (FC) | Soil | mm | 0–600 |
| Nonlinearity parameter (BETA) | Soil | | 0–20 |
| Very fast storage coefficient (k0) | Runoff | days | 0–2 |
| Fast storage coefficient (k1) | Runoff | days | 2–30 |
| Slow storage coefficient (k2) | Runoff | days | 30–250 |
| Upper storage coefficient (Lsuz) | Runoff | mm | 1–100 |
| Percolation rate (Cperc) | Routing | mm/day | 0–8 |
| Maximum base parameter (Bmax) | Routing | days | 0–30 |
| Free scaling parameter (Croute) | Routing | day2/mm | 0–50 |

Model calibrations are additionally evaluated by the runoff volume error (VE) and the root mean square error (RMSE) between measured and simulated snow water equivalents. VE is the measure of bias between the simulated and observed runoff [22]. The VE value equal to 0 indicates no bias, VE smaller than 0 represents an underestimation of the total runoff volume, and VE greater than 0 denotes overestimation of the total runoff volume. The equation is defined as:

$$VE = \frac{\sum\limits_{i=1}^{n} Q_{sim,i} - \sum\limits_{i=1}^{n} Q_{obs,i}}{\sum\limits_{i=1}^{n} Q_{obs,i}} \tag{4}$$

The root mean square error (RMSE) between simulated and observed daily SWE values is calculated as:

$$RMSE = \sqrt{\sum\limits_{i=1}^{n} \frac{(SWE_{sim,i} - SWE_{obs,i})^2}{n}} \tag{5}$$

$SWE_{sim,i}$ and $SWE_{obs,i}$ are simulated and observed SWE values on day $i$, respectively, $n$ is the number of observations. Smaller RMSE values indicate better agreement between the observed and simulated SWE.

Model performance is evaluated by the differential split-sample test [44], where the model calibrated in one of the six years long periods is validated in the remaining (four) periods.

*3.2. Analysis of Uncertainty Resulting from a Different Number of Calibration Repetitions*

The model is calibrated automatically 500 times in each time period. The effect of different numbers of calibration repetitions on model performance and parameter uncertainty is examined for five groups of calibration repetitions, i.e., first 10, 50, 100, 300, and finally, all 500 repetitions.

Our experiment looks at the repetitions of the calibrations of a hydrologic model, where the calibration strategy is identical (no difference in the number of calibration runs). The only difference is that the optimisation approach is based on a random generation of the initial population of model parameters, which can have some impact on the final result (i.e., calibrated parameter set). We have thus run the entire calibration strategy 500 times and investigated the impact on different model efficiencies (objective function, its parts and efficiencies not used in objective function).

## 4. Results

### 4.1. Uncertainty of Hydrologic Model Performance in Varying Climatic Conditions

Figure 3 shows the variability in the medians and ranges of the objective function (OF), the logarithmic Nash-Sutcliffe efficiency (logNSE), the Nash-Sutcliffe efficiency (NSE), volume error (VE) and the root mean square error (RMSE) for a different number of calibration repetitions (i.e., 10, 50, 100, 300 and 500 repetitions) in five specific calibration periods. The results show that the median difference between the groups of calibration repetitions was very small even for snow RMSE, which was not used in model calibration. The variability in the range of model efficiency was, however, larger. The differences among the groups were larger for VE and RMSE. It indicates that an increasing number of calibration repetitions increased also the variability of model simulations. Hence, the spread of the efficiency criteria increased as well, particularly of those criteria that were not included in model calibration (runoff VE and snow RMSE). Interestingly, the absolute value of the VE and RMSE ranges differed, but the difference between 10 and 500 calibration repetitions did not vary among the climate periods. While larger VE range was observed in drier periods, the largest snow RMSE range for most of the calibration groups was found in the recent period (2013–2018) that had many snow poor winters with shallower snowpacks and on average smaller SWE maximum (Table 1). Generally, the lowest model performance was obtained in the period 1989–1994, which was the warmest and comparatively dry. The largest efficiency in terms of the objective function, NSE and snow RMSE was obtained in the most recent period 2013–2018, which was second warmest and driest with second smallest average SWEmax. The volume error of model simulations in that period, however, indicated more than 10% underestimation of runoff volumes.

Figures 4 and 5 show the variability in the median (Figure 4) and ranges (Figure 5) of model efficiencies in four specific validation periods, i.e., parameters obtained in each calibration period were validated in the remaining four validation periods. Similarly, as for the calibration model efficiency, the medians in most cases did not change with the number of calibration repetitions. The exception was obtained for the validation efficiency in the coldest period when the model was calibrated in the warmest period (the first column of panels in Figure 4). Medians of runoff objective function and logNSE efficiency for 10 calibration repetitions were greater than the median for groups with a higher number of calibration repetitions. The same pattern was reflected also in the range of the model efficiencies (Figure 5). The variability in model efficiency was the lowest for 10 calibration repetitions and it was increasing with the number of calibration repetitions. The variability was larger for the efficiency criteria that were not used in model calibration (VE and snow RMSE). Larger ranges in the efficiency criteria during model validation were found when the model was validated in colder time periods (i.e., 1995–2000 and 2001–2006). For snow simulations, the calibration in the snow poorer period (2013–2018) lead to greater variability of RMSE if the model was validated in periods with larger snowpacks (1995–2000).

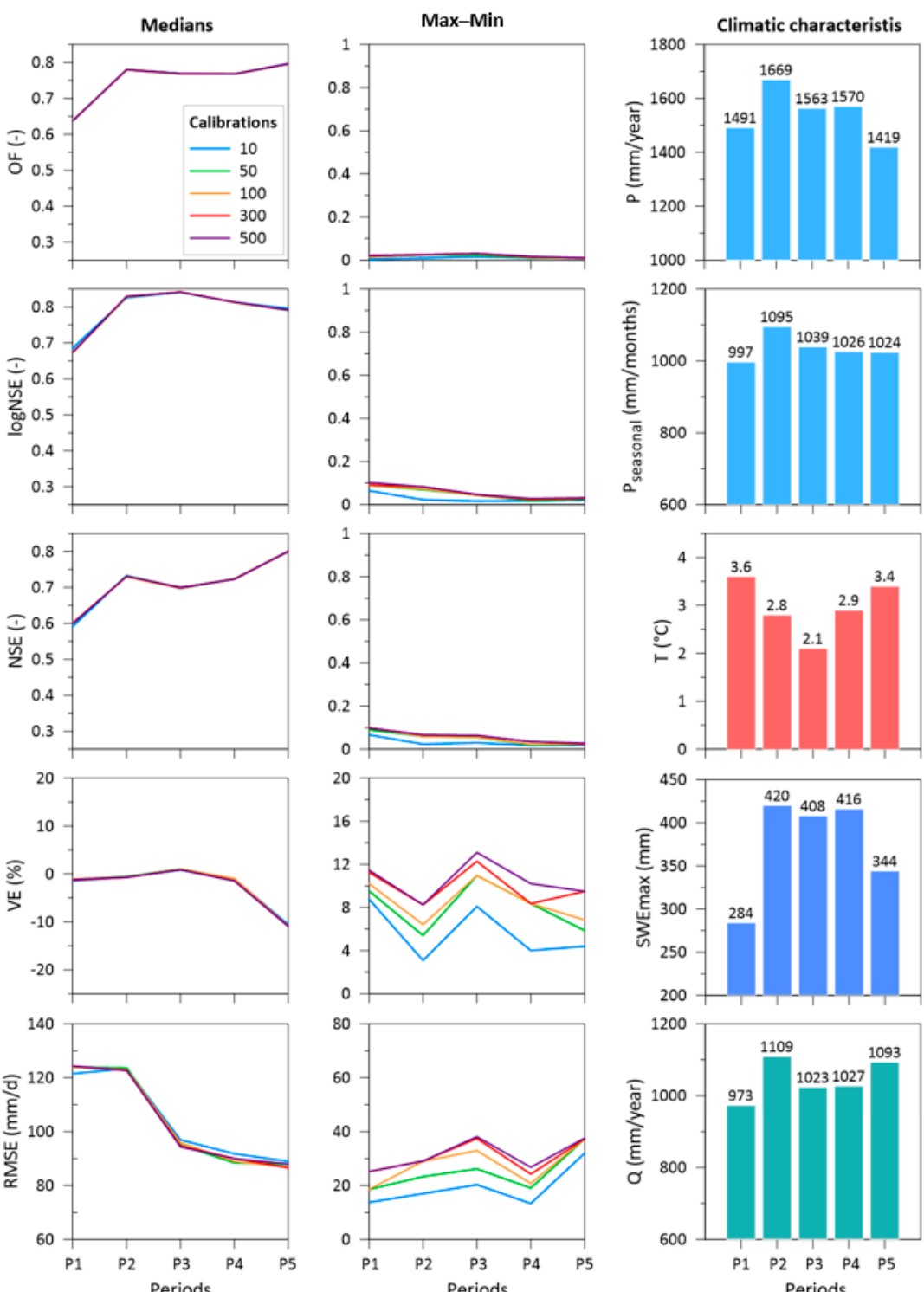

**Figure 3.** Variability of the objective function (OF), the logarithmic Nash-Sutcliffe efficiency (logNSE), the Nash-Sutcliffe efficiency (NSE), the root mean square error (RMSE) and the volume error (VE) after 10, 50, 100, 300, 500 calibration repetitions in five specific periods. The lines represent the medians and ranges (max–min) of different metrics obtained by different calibration repetitions. The right panel shows the variability of climatic characteristics in the five periods.

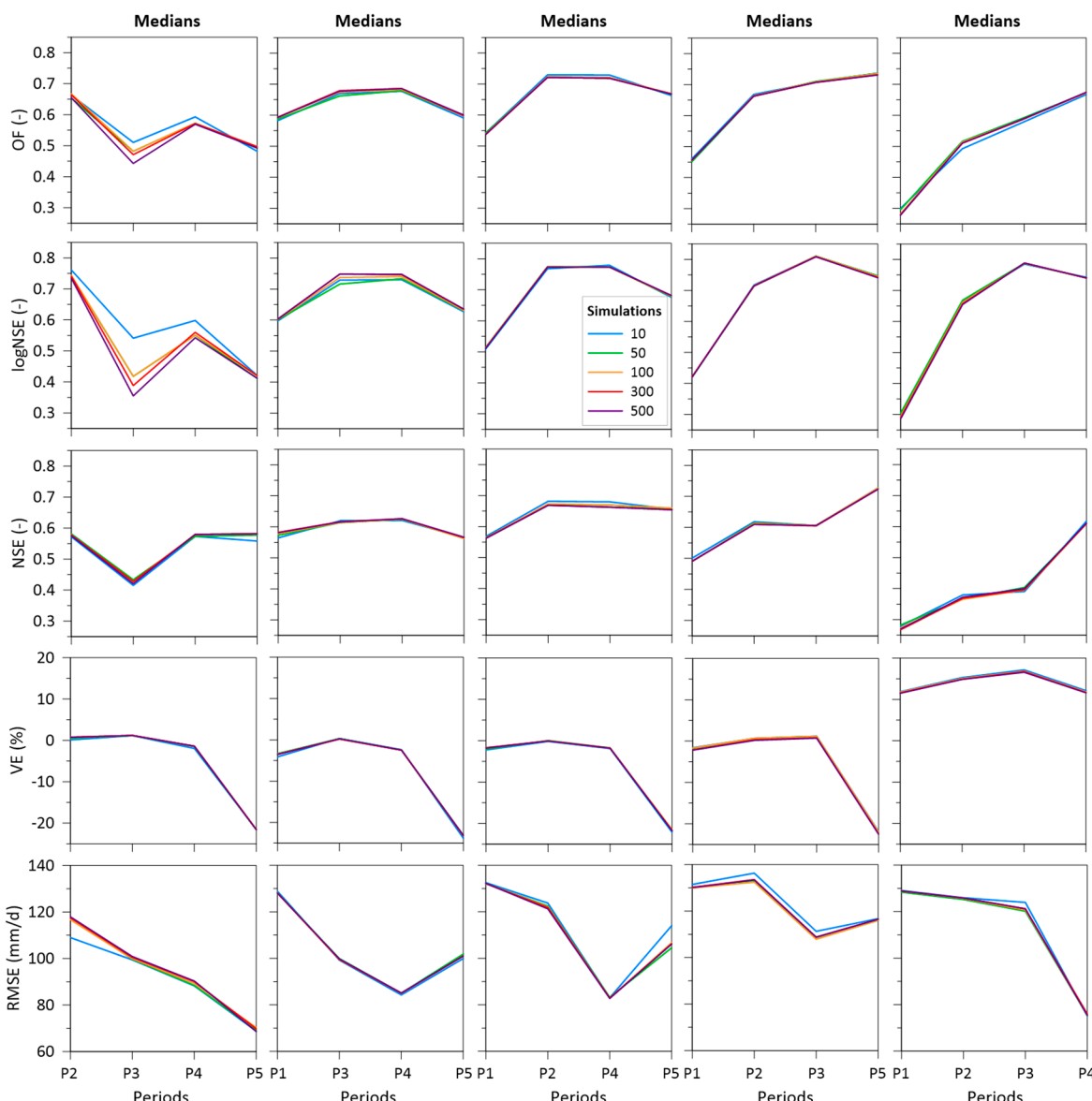

**Figure 4.** Variability of medians of the objective function (OF), the logarithmic Nash-Sutcliffe efficiency (logNSE), the Nash-Sutcliffe efficiency (NSE) criteria, the root mean square error (RMSE) and the volume error (VE) after 10, 50, 100, 300, 500 repetitions of model simulations in four validation periods. The lines represent the medians of different metrics obtained by the different number of repetitions.

### 4.2. Uncertainty in Hydrologic Model Parameters in Varying Climatic Conditions

The differences in the median of model parameters and their variability are plotted in Figures 6 and 7. The panels show parameters of the main model modules, i.e., the snow, soil moisture, runoff generation and runoff routine modules. The results indicate that the median of model parameters varied among the time periods but generally did not change much among the calibration groups. Increasing the number of calibration repetitions did not result in changes in the median values for most model parameters. The exceptions include routing module parameters that were generally less sensitive to selected model efficiencies; the snow degree-day parameter calibrated in the coldest time period and the limit for potential evapotranspiration Lprat and very fast storage parameter k0 calibrated in the warmest time period (P1). Medians of these parameters for the calibration group of 10 repetitions were noticeably different than the median obtained from greater numbers of calibration repetitions.

Significantly larger differences among calibration groups were found for the ranges of model parameters (Figure 7) that were increasing with the number of calibration repetitions. Such a pattern was consistent over all calibration periods. There were also large differences between ranges in model parameters calibrated in different time periods. While the difference in ranges of snow module parameters was the smallest for the snow correction factor (SCF), the less sensitive threshold temperatures (Tr, Ts, Tm) have the largest variability among the calibration groups. The largest differences among the calibration groups for degree-day factor (DDF) were found in the warmest period P1 (1989–1994). Soil moisture and runoff generation parameters had the largest differences in the wettest period P2 (1995–2000), where the ranges obtained for 10 calibration repetitions were significantly smaller than those obtained for a greater number of calibration repetitions.

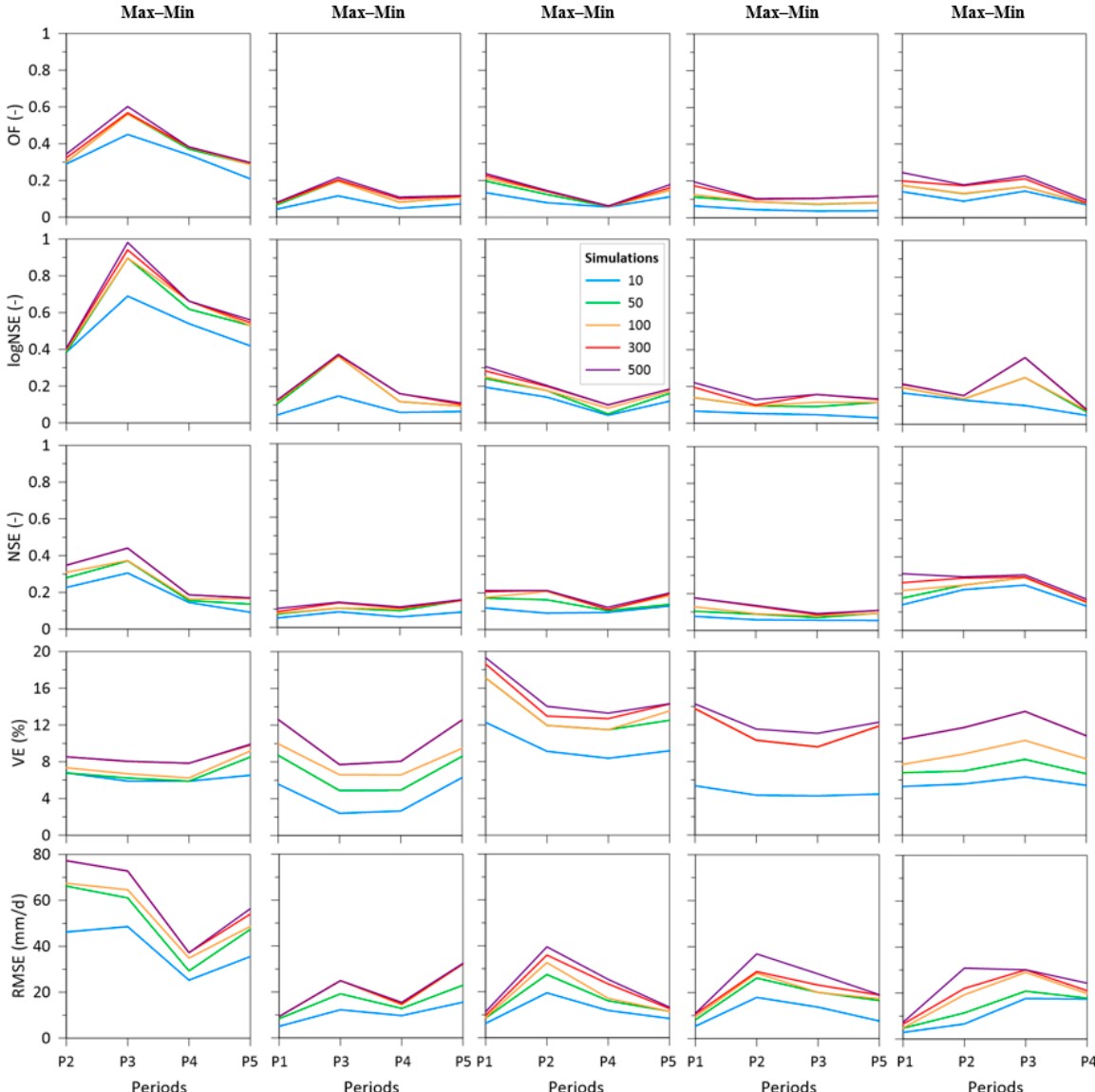

**Figure 5.** Variability of the ranges of the objective function (OF), the logarithmic Nash-Sutcliffe efficiency (logNSE), the Nash-Sutcliffe efficiency (NSE) criteria, the root mean square error (RMSE) and the volume error (VE) after 10, 50, 100, 300, 500 repetitions of model simulations in four validation periods. The lines represent the ranges (Max–Min) of different metrics obtained by the different number of repetitions.

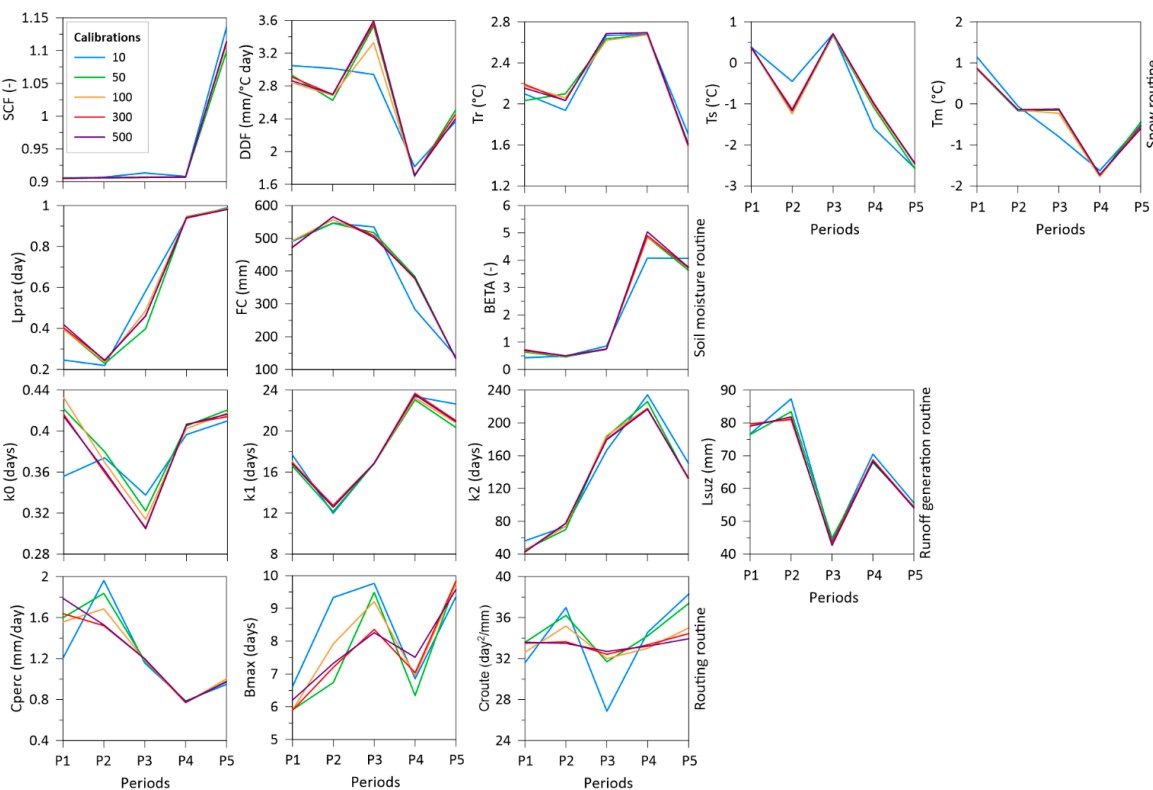

**Figure 6.** Medians of model parameters after 10, 50, 100, 300, 500 calibration repetitions in five specific periods.

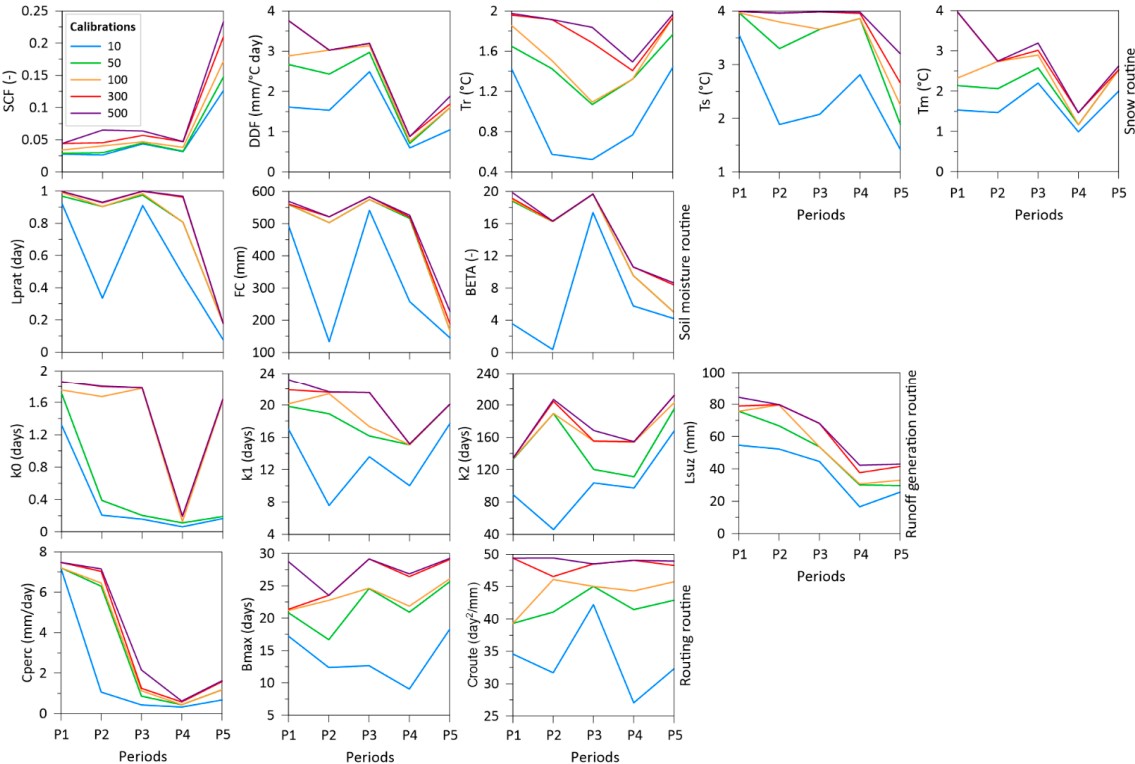

**Figure 7.** Ranges of model parameters after 10, 50, 100, 300, 500 calibration repetitions in five specific periods.

### 4.3. Simulation of Runoff and Snow Water Equivalent in Varying Climatic Conditions

In order to demonstrate the implications of increasing variability in model parameters caused by the increasing number of calibration repetitions, Figure 8 compares ranges of model simulations obtained by 10 and 500 simulations with observed runoff values. Runoff is simulated in a colder/wetter validation period (2001–2006) with model parameters obtained in a warmer/drier calibration period (1989–1994). Better simulations are indicated by smaller variability between min. and max (i.e., the lines are closer to each other).

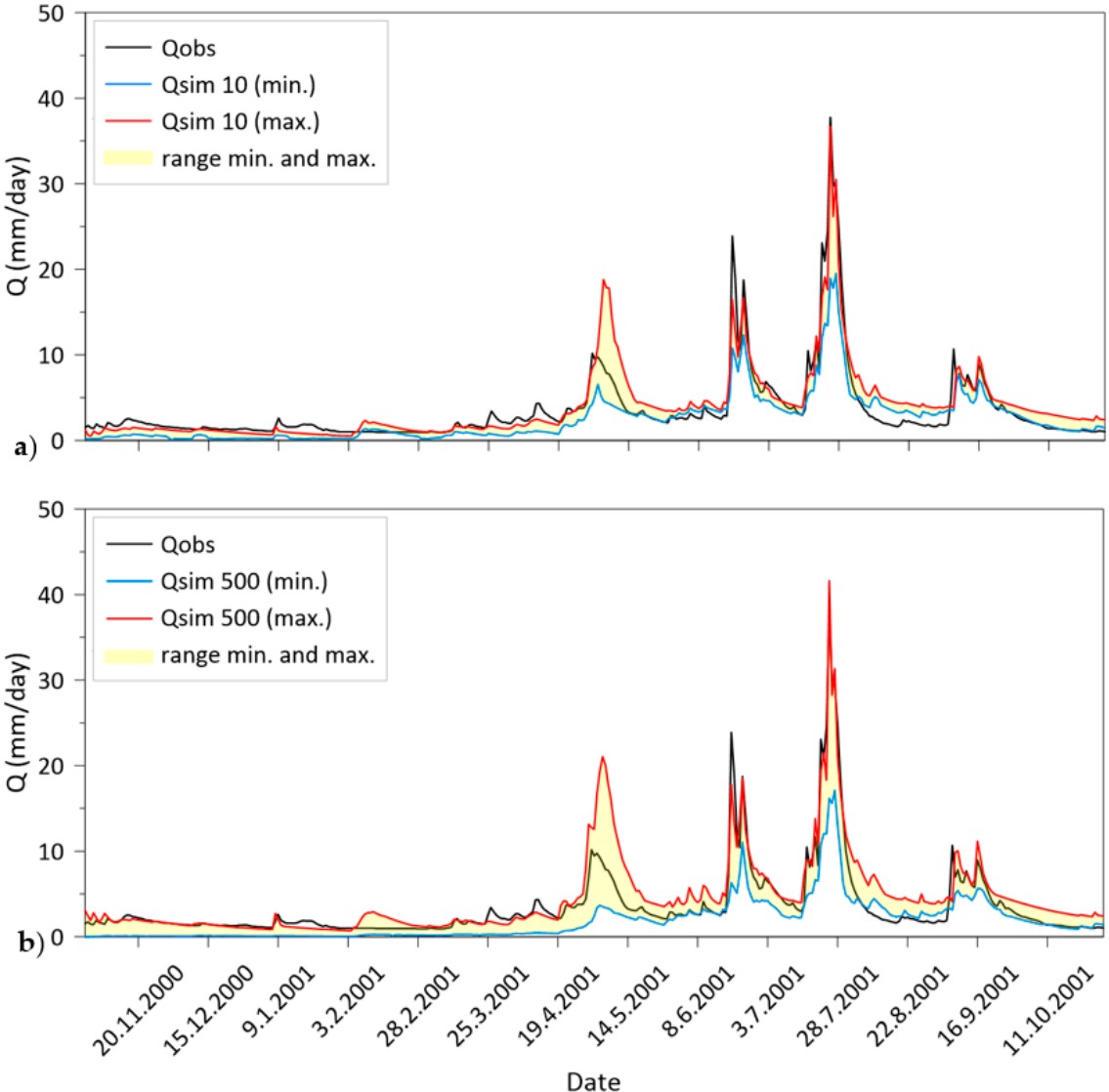

**Figure 8.** Comparison of measured and simulated runoff after 10 (**a**) and 500 simulations (**b**). Runoff is simulated in a colder/wetter validation period (2001–2006) with model parameters obtained in a warmer/drier calibration period (1989–1994). The yellow colour indicates the range between the minimum (blue line) and maximum (red line) simulated runoff obtained by 10 and 500 repetitions.

For a detailed illustration of the results, one hydrological year (1, November 2000 to 1, October 2001) is plotted (Figure 8). Figure 8 shows that the model reproduced runoff variability. Slightly more realistic simulations of runoff volume, mainly during summer months, were obtained using a smaller number of simulations (i.e., with 10 repetitions of model simulations). Greater differences between simulated and measured runoff were related to the main snowmelt period (April, May).

Figure 9 presents measured and simulated snow water equivalent (SWE) after 10 and 500 simulations. While measured data are represented by point (snow course) measurements of SWE at site Červenec (catchment mean elevation 1500 m a.s.l.), the simulations represent catchment SWE. The simulations were based on model parameters obtained in the snow poorer calibration period (1989–1994). We plotted one hydrological year (1, November 2000 to 1, October 2001) in the snow richer validation period (2001–2006). The range of simulated SWE based on 10 simulations was slightly smaller than that based on 500 simulations. In some cases (e.g., 15 February 2001 to 9 March 2001) the simulated catchment mean SWE was smaller than SWE measured at catchment mean altitude. Such a result could be realistic if the SWE significantly increases with the altitude.

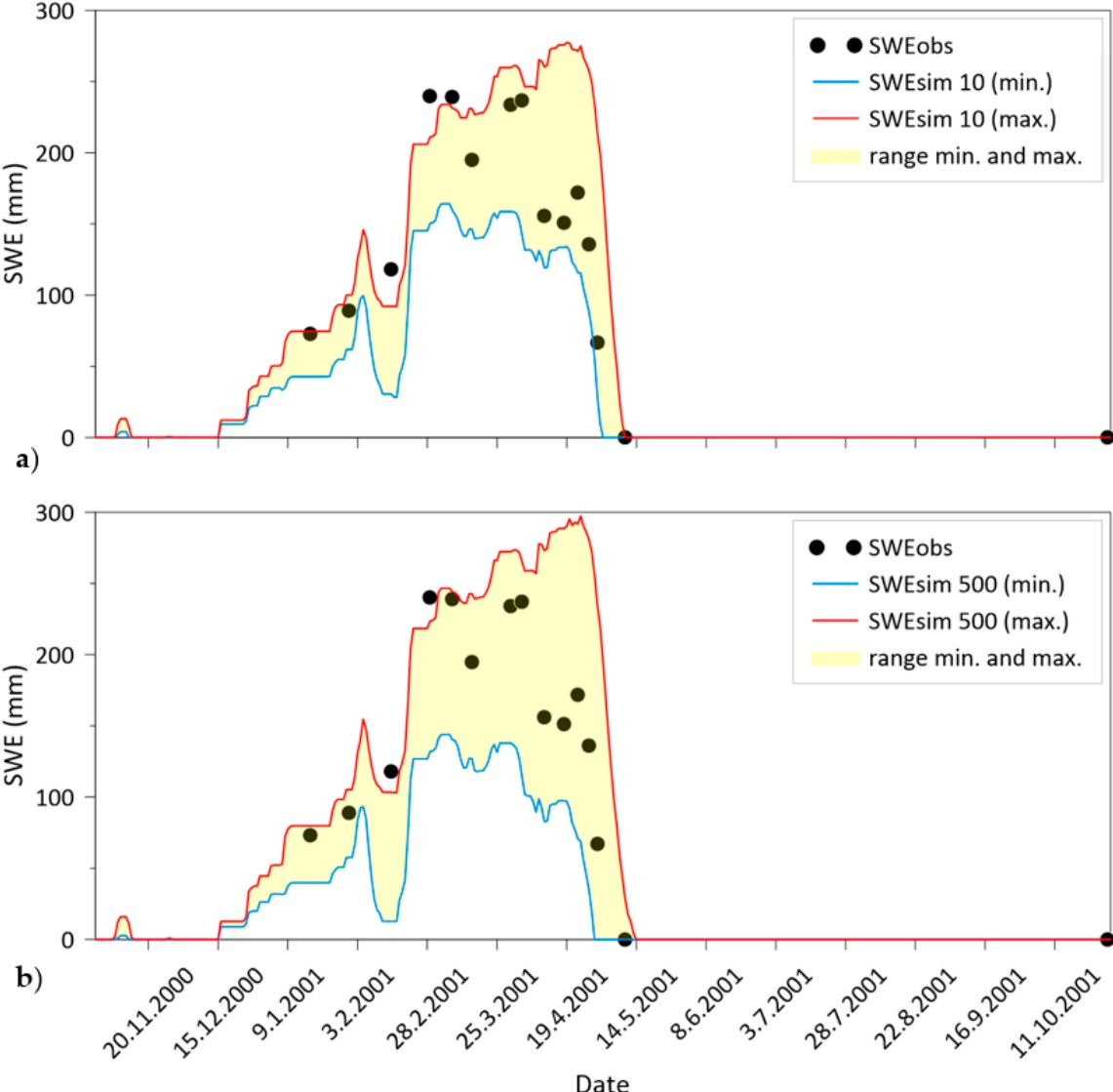

**Figure 9.** Measured snow water equivalent (SWE) at catchment mean elevation (points) and simulated catchment SWE after 10 (top panel) (**a**) and 500 repetitions (**b**). SWE is simulated in a snow richer validation period (2001–2006) with model parameters obtained in the snow poorer calibration period (1989–1994). The blue and red colours indicate min. and max. SWE values obtained by 10 and 500 simulations, respectively.

## 5. Discussion

This study evaluates the impact of a different number of calibration repetitions on model performance and parameter uncertainty in varying climate periods observed in a small pristine mountain catchment. Previous studies showed that an increasing number of calibration runs can be the way to the more robust calibration of conceptual hydrologic models. For example, Finger et al. [9] used the Monte Carlo (MC) approach and found that 100 ensemble parameter sets (out of 10,000 MC runs) are sufficient to obtain an adequate model performance and parameter variability. A similar conclusion is presented in Finger et al. [10] and Konz and Seibert [8] for alpine catchments in Austria and Switzerland. Another example is the results of one experiment of Ceola et al. [36] who compared model performance and uncertainty for 10 and 100 calibration runs. They found some differences in the calibrated model parameters which have been interpreted as a result of parameter insensitivity, equifinality and the problem of model structure to capture the complex runoff generation processes in some catchments. Our results indicate that 10 calibration repetitions resulted in the same model performance, but a smaller range of model parameters and model efficiency and hence model uncertainty compared to a larger number of calibration repetitions. The variability in model performance tends to increase with the increasing number of calibration repetitions, particularly for validation periods and efficiency measures which were not used in model calibration, i.e., in our case runoff volume error and snow modelling error.

Our study extends previous assessments in terms of investigating whether varying climate conditions have an additional impact on the performance of a hydrological model and uncertainty of results evaluated on the basis of a different number of calibration repetitions. Previous studies clearly reported that the performance of conceptual hydrologic models tends to degrade when climatic conditions of model simulations are different from those used for model calibration. For example, Merz et al. [22] showed that the model calibrated in a colder decade had a tendency to overestimate the runoff in a warmer decade and vice versa. Similarly, Coron et al. [23] showed that model calibration in wetter conditions leads to runoff overestimation in drier periods. These results were attributed to the impact of increasing air temperature on the main runoff generation mechanisms, particularly to increasing nonlinearity in runoff response in warmer and drier climate conditions. We found that although the absolute value of model efficiency and parameter ranges varies between the different climate periods, the difference in model performance and parameter variability between 10 and 500 calibration repetitions is generally consistent among all tested periods. The exceptions are larger differences in soil and runoff generation model parameters which are calibrated in the wettest time period. Here the increasing number of calibration repetitions increased the variability in model parameter ranges. For snow model parameters, we found similar results as presented in Merz et al. [22] and Sleziak et al. [34]. Model parameters calibrated in warmer periods resulted in greater differences among the different calibration groups, which could be attributed to the impact of increasing air temperatures on the variability of snowmelt.

## 6. Conclusions

We explored some uncertainties associated with the calibration of a lumped hydrological model in a small mountain research catchment. The results showed that a relatively small number of 10 calibration repetitions in model parameters optimisation can provide robust model simulations with smaller parameter uncertainty than obtained by a larger number of calibration repetitions. Model performance based on ten calibration repetitions was particularly better when model parameters were optimised in colder/wetter climate conditions.

To use these findings in practice (e.g., when dealing with various water management tasks), it should be noted that the selection of the optimal set of parameters also depends on the type of the task to be addressed (e.g., some sets of parameters may better describe periods of low flow while other can better represent high flows). The results are also dependent on the model used.

More analysis needs to be done in the future to verify these results. In our next effort, we plan to investigate modelling uncertainties in the context of climate change by using distributed hydrological models.

**Author Contributions:** Conceptualization, P.S., J.P. and L.H.; methodology, P.S., L.H. and J.P.; software, P.S.; validation, P.S.; formal analysis, P.S., J.P., L.H. and M.D.; investigation, P.S., L.H., J.P. and M.D.; writing—original draft preparation, P.S., J.P. and L.H.; writing—review and editing, P.S., J.P. and L.H.; visualization, P.S. All authors have read and agreed to the published version of the manuscript.

**Funding:** The study was supported by grants from the Slovak Academy of Sciences VEGA (project No. 2/0065/19) and the Slovak Research and Development Agency (APVV) (project No. 15-0497). Data collection was supported by project ITMS 26210120009 Infrastructure completion of hydrological research stations, of the Research & Development Operational Programme funded by the ERDF.

**Acknowledgments:** We would like to acknowledge the support from the Stefan Schwarz grant of the Slovak Academy of Sciences given to the first author.

**Conflicts of Interest:** The authors declare no conflict of interest.

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
