# Peer review of "Uncertainty in the Number of Calibration Repetitions of a Hydrologic Model in Varying Climatic Conditions"

_water, doi:10.3390/w12092362_

Round 1
Reviewer 1 Report
This paper deals with the impact of a different number of calibration runs on hydrologic model performance and parameter uncertainty in varying climatic conditions.
Authors used Data, for the period 1989 – 2018, from An alpine catchment in the Western Tatra Mountains .
A lumped conceptual hydrologic model is calibrated by an automatic optimization using a differential evolution algorithm approach in order to evaluate the impact of a different number of calibration runs on model performance and parameter uncertainty in varying climate periods observed in the small catchment.
The novelty of this study is the intention to extend previous assessments in terms of investigating whether varying climate conditions have some additional impact on the performance and uncertainty of a different number of calibration runs. Previous studies clearly reported that the performance of conceptual hydrologic models tends to degrade when climatic conditions of model simulations are different from those used for model calibration.
I agree with the authors when they said, in Conclusion section, “To use these findings in practice (e.g., when dealing with various water management tasks), it should be noted that the selection of the optimal set of parameters also depends on the type of the task to be addressed (e.g., some set of parameters may better describe periods of low flow, other parameters may better represent high flows). The results are also dependent on the model used.”
It would be interesting if authors could investigate, also, how their results can vary, if some variation will appear as well, by assuming as measures not used in calibration (the runoff volume error and the root mean square error of snow simulations) other measures.
I suggest, also, to improve the readability of figures 3-7.
Author Response
Reviewer #1:
This paper deals with the impact of a different number of calibration runs on hydrologic model performance and parameter uncertainty in varying climatic conditions.
Authors used Data, for the period 1989 – 2018, from an alpine catchment in the Western Tatra Mountains.
A lumped conceptual hydrologic model is calibrated by an automatic optimization using a differential evolution algorithm approach in order to evaluate the impact of a different number of calibration runs on model performance and parameter uncertainty in varying climate periods observed in the small catchment.
The novelty of this study is the intention to extend previous assessments in terms of investigating whether varying climate conditions have some additional impact on the performance and uncertainty of a different number of calibration runs. Previous studies clearly reported that the performance of conceptual hydrologic models tends to degrade when climatic conditions of model simulations are different from those used for model calibration.
I agree with the authors when they said, in Conclusion section, “To use these findings in practice (e.g., when dealing with various water management tasks), it should be noted that the selection of the optimal set of parameters also depends on the type of the task to be addressed (e.g., some set of parameters may better describe periods of low flow, other parameters may better represent high flows). The results are also dependent on the model used.”
Response: We would like to thank the reviewer for evaluation of the manuscript. Our responses are given below. We have also revised English and checked grammar and spelling using a tool grammarly (https://www.grammarly.com/).
Comment 1: It would be interesting if authors could investigate, also, how their results can vary, if some variation will appear as well, by assuming as measures not used in calibration (the runoff volume error and the root mean square error of snow simulations) other measures.
Response: Thank you for this comment. More detailed assessment would be interesting, but it goes beyond the scope of this study. We plan to analyze it more deeply in our future work. As an example here we attach a figure (see Word file) showing the variability in the median and ranges of the low flows (Q95) for a different number of calibration repetitions (i.e., 10, 50, 100, 300 and 500 repetitions) in five specific calibration periods.
Comment 2: I suggest, also, to improve the readability of figures 3-7.
Response: Thank you for the comment. In response to this comment we have changed the thickness of the lines (see Figures in full resolution in zip file, or in Word manuscript).

Reviewer 2 Report
The authors analyzed the impact of the number of calibration runs on the performance and its range using a HBV type conceptual hydrologic model. The topic is practically important as any modelers should face this problem. However, I have some concerns in the design of the analysis and, more importantly, in the result itself. In spite of having contradictory results from previous studies, i.e. smaller number of calibration runs leads to smaller variability, the authors did not explain the reason for this. Including this, the authors should address the following comments before the publication of this manuscript.
Overall
It is surprising that smaller number of calibration runs results in smaller performance and parameter value variability. This is hard to understand because, if optimization processes go reasonably, as the number of runs increases the parameters should get to more optimal values and thus into a smaller range. Otherwise, optimization did not go successfully. Please make a reasonable explanation in result/discussion section, which must be addressed to make this manuscript acceptable for publication.
Overall
Related to the above comment, I think that smaller number cases might be more affected by initial values. Please make discussions on the impact of initial parameter values (values for the first calibration run) on performance and parameter value range.
Overall
Design of uncertainty analysis in calibration numbers is not clear.
In case of 100 runs, 100-run calibration was repeated 500 times, or 5 sets of 100 runs were extracted from 500 runs (i.e. all the analyses were based on single 500-run calibration)? If latter, how to get max and min of 300- and 500-run cases?
Title
The title makes readers imagine that integrated uncertainty analysis, i.e. multiple uncertainty sources in calibration processes are treated in the study; however, as clearly summarized in the first sentence of "5. Discussion", the research actually analyzed a more specific component "the number of calibrations in the optimization procedure". The title should be more specified such as: Uncertainty in the number of calibration runs of a hydrologic model in varying climatic conditions. Since the discussions and conclusions derived from the analysis are not specific to a HBV type model, "a HBV type conceptual model" could be removed if the title is too long.
l. 124
1984-1989 is mistake of 1988-1994? I don't find period 1984-1989.
Fig. 2
Period 5 (2013-2018) is a warmer period wih smaller precipitation, but it has the largest runoff from April to September (black line). Could the authors explain why?
3.1 Hydrologic model
Please make a summary table or conceptual figure of the model because it is hard to understand the relationship and roles of 15 model parameters at glance, which leads to huge confusion in the interpretation of Figs. 6 and 7 though the layout of panels are still considerate.
l.166
Explain why logNSE was used in addition to NSE. For checking performance of low flows?
Fig. 3
All the performance indices look better for later periods. Please explain why this happens in the manuscript. The calibration was not conducted independently among the periods?
Figs. 5 and 7
(max. - min.) should be the rate to the median because it is hard to interpret the scale of variability from the absolue difference.
l. 276
The expression of the date is strange. Make it such as "1, Nov., 2000 to 1, Oct., 2001" or "Nov. 1, 2000 to Oct. 1, 2001"
Author Response
Reviewer #2:
The authors analyzed the impact of the number of calibration runs on the performance and its range using a HBV type conceptual hydrologic model. The topic is practically important as any modelers should face this problem. However, I have some concerns in the design of the analysis and, more importantly, in the result itself. In spite of having contradictory results from previous studies, i.e. smaller number of calibration runs leads to smaller variability, the authors did not explain the reason for this. Including this, the authors should address the following comments before the publication of this manuscript.
Response: We would like to thank the reviewer for a thorough and careful evaluation, and constructive and very helpful comments on the manuscript. We have considered and accepted all comments and suggestions for revision. We made the changes using track changes. Track changes are shown in a different font colour and formatted with underlines (insertions) or strikethroughs (deletions). The specific changes are highlighted in yellow. We have also revised English and checked grammar and spelling using a tool grammarly (https://www.grammarly.com/).
Reading the comment, we realized that we have probably not described the design clearly. In the revised manuscript we have attempted to do it. Our experiment looks on the repetitions of the calibrations of a hydrologic model, where calibration strategy is identical (no difference in the number of calibration runs). The only difference is that the optimization approach is based on random generation of the initial population of model parameters, which can have some impact on the final result (i.e. calibrated parameter set). We have thus ran the entire calibration strategy 500 times and investigated the impact on different model efficiencies (objective function, its parts and efficiencies not used in objective function). In response to this comment we have revised the description of the experiment design (lines 210–215 with track changes; lines 195–200 without track changes; changes are highlighted in yellow). In order to clarify text, we have changed “calibration runs” to “calibration repetitions”. These changes are also highlighted in yellow.
Comment 1: It is surprising that smaller number of calibration runs results in smaller performance and parameter value variability. This is hard to understand because, if optimization processes go reasonably, as the number of runs increases the parameters should get to more optimal values and thus into a smaller range. Otherwise, optimization did not go successfully. Please make a reasonable explanation in result/discussion section, which must be addressed to make this manuscript acceptable for publication.
Response: Thank you for the comment. Please see the comment above. In response to this comment we have revised the description of our experiment. (lines 210–215 with track changes; lines 195–200 without track changes; changes are highlighted in yellow).
Comment 2: Related to the above comment, I think that smaller number cases might be more affected by initial values. Please make discussions on the impact of initial parameter values (values for the first calibration run) on performance and parameter value range.
Response: Thank you for the comment. Please see the comment above. In response to this comment we have revised the description of our experiment. (lines 210–215 with track changes; lines 195–200 without track changes; changes are highlighted in yellow)
Comment 3: Design of uncertainty analysis in calibration numbers is not clear.
In case of 100 runs, 100-run calibration was repeated 500 times, or 5 sets of 100 runs were extracted from 500 runs (i.e. all the analyses were based on single 500-run calibration)? If latter, how to get max and min of 300- and 500-run cases?
Response: We have improved the description of the methodology used in the study. (lines 210–215 with track changes; lines 195–200 without track changes; changes are highlighted in yellow). In fact, we did 500 calibrations and then compared results based on the first 10, 50, 100, 300 (and 500) calibrations.
Comment 4: The title makes readers imagine that integrated uncertainty analysis, i.e. multiple uncertainty sources in calibration processes are treated in the study; however, as clearly summarized in the first sentence of "5. Discussion", the research actually analyzed a more specific component "the number of calibrations in the optimization procedure". The title should be more specified such as: Uncertainty in the number of calibration runs of a hydrologic model in varying climatic conditions. Since the discussions and conclusions derived from the analysis are not specific to a HBV type model, "a HBV type conceptual model" could be removed if the title is too long.
Response: Thank you for the comment. The tittle has been modified as suggested by the reviewer. The title now reads: “Uncertainty in the number of calibration repetitions of a hydrologic model in varying climatic conditions”.
Comment 5: l. 124: 1984-1989 is mistake of 1988-1994? I don't find period 1984-1989.
Response: Thank you this mistake has been corrected. (line 135 with track changes; line 124 without track changes; highlighted in yellow)
Comment 6: Fig. 2: Period 5 (2013-2018) is a warmer period with smaller precipitation, but it has the largest runoff from April to September (black line). Could the authors explain why?
Response: A recent assessment of changes in hydrological cycle of the Jalovecký Creek catchment indicated greater dynamics (i.e., higher runoff coefficients, number of flow reversals per year or flashiness index, greatest annual and seasonal discharge) in the last years 2014–2018 (Holko et al., 2020a, b). Higher variability of runoff in the most recent period 2013-2018 was also identified in Sleziak et al. (2019). We are not yet sure about the reasons. The attribution analysis pointed at changes in precipitation regime, but the correlations explained a relatively small part of variability. It is possible that other factors like changes in vegetation cover or in catchment water storage also played some role.
Holko, L.; Sleziak, P.; Danko, M.; Bičárová, S.; Pociask-Karteczka, J. Analysis of changes in the hydrological cycle of a pristine mountain catchment. 1. Water balance components and snow cover. J. Hydrol. Hydromech. 2020a, 68, 2, 180–191. doi: 10.2478/johh-2020-0010.
Holko, L.; Danko, M., Sleziak, P. Analysis of changes in the hydrological cycle of a pristine mountain catchment. 2. Isotopic data, trend and attribution analyses. J. Hydrol. Hydromech. 2020b, 68, 2, 192–199. DOI: 10.2478/johh-2020-0011.
Sleziak, P.; Danko, M.; Holko, L. Testing an alternative approach to calibration of a hydrological model under varying climatic conditions. Acta Hydrologica Slovaca 2019, 20, 131-138, doi: 10.31577/ahs-2019-0020.02.0015.
Comment 7: 3.1 Hydrologic model: Please make a summary table or conceptual figure of the model because it is hard to understand the relationship and roles of 15 model parameters at glance, which leads to huge confusion in the interpretation of Figs. 6 and 7 though the layout of panels are still considerate.
Response: Thank you for the comment. Table regarding 15 TUW model parameters has now been included in the manuscript. Please find it in Subsection 3.1. Hydrologic model.
Comment 8: l.166: Explain why logNSE was used in addition to NSE. For checking performance of low flows?
Response: The objective function (i.e., the combination of NSE and logNSE) used in calibration is selected on the basis of prior analyses performed in different calibration studies (see e.g. Sleziak et al., 2018, 2019). We used both metrics (i.e., NSE and logNSE) in order to achieve a more balanced evaluation of flows.
Sleziak, P.; Szolgay, J.; Hlavčová, K.; Duethmann, D.; Parajka, J.; Danko, M. Factors controlling alterations in the performance of a runoff model in changing climate conditions. J. Hydrol. Hydromech. 2018, 66, 381–392, https://doi.org/10.2478/johh-2018-0031.
Sleziak, P.; Danko, M.; Holko, L. Testing an alternative approach to calibration of a hydrological model under varying climatic conditions. Acta Hydrologica Slovaca 2019, 20, 131-138, doi: 10.31577/ahs-2019-0020.02.0015.
Comment 9: Fig. 3: All the performance indices look better for later periods. Please explain why this happens in the manuscript. The calibration was not conducted independently among the periods?
Response: The calibration was conducted independently among the studied periods. Our results indicate that the model performance is larger when the model parameters are optimized in colder/wetter climate conditions. All evaluated metrics are higher in more recent periods, instead of volume errors (VE), which show opposite patterns (i.e., VE are always significantly lower than zero that represents an underestimation of simulated runoff.). It could be caused by higher variability of flows which was identified in Sleziak et al. (2019).
Sleziak, P.; Danko, M.; Holko, L. Testing an alternative approach to calibration of a hydrological model under varying climatic conditions. Acta Hydrologica Slovaca 2019, 20, 131-138, doi: 10.31577/ahs-2019-0020.02.0015.
Comment 10: Figs. 5 and 7: (max. - min.) should be the rate to the median because it is hard to interpret the scale of variability from the absolute difference.
Response: Thank you for the comment. While the median differences between the groups of calibration/simulation repetitions are very small (even for metrics, which are not used in model calibration), the ranges (max. - min.) show a larger variability. Our intention is to show larger differences among the groups of calibration/simulation repetitions that are later visible in comparison of measured and simulated runoff/snow water equivalent (Figures 8 and 9).
Comment 11: l. 276: The expression of the date is strange. Make it such as "1, Nov., 2000 to 1, Oct., 2001" or "Nov. 1, 2000 to Oct. 1, 2001"
Response: This has been corrected. (lines 314 and 331 with track changes; lines 289 and 303 without track changes; highlighted in yellow)

Round 2
Reviewer 2 Report
I think all of my comments were addressed. It is interesting that calibration in a specific climatic (wetter/colder) condition makes better performance.
I leave two comments as reference. Please consider for the next step.
1) I would like the authors to explore the interpretation for the above finding in the next research.
2) Previous studies showed larger number of repititions is an important way for robust model calibration, but this study suggests that only 10 repititions show comparable performance with 500 ones while the former shows smaller parameter ranges; is there possibility that calibration in this study was easider and thus each calibration among smaller number (10) of repititions was similar to each other?